# MSCs as Tumor-Specific Vectors for the Delivery of Anticancer Agents—A Potential Therapeutic Strategy in Cancer Diseases: Perspectives for Quinazoline Derivatives

**DOI:** 10.3390/ijms23052745

**Published:** 2022-03-02

**Authors:** Monika Szewc, Elżbieta Radzikowska-Bűchner, Paulina Wdowiak, Joanna Kozak, Piotr Kuszta, Ewa Niezabitowska, Joanna Matysiak, Konrad Kubiński, Maciej Masłyk

**Affiliations:** 1Department of Human Anatomy, Medical University of Lublin, 20-090 Lublin, Poland; paulina.wdowiak@umlub.pl (P.W.); joanna.kozak@umlub.pl (J.K.); pkuszta@gmail.com (P.K.); 2Department of Plastic, Reconstructive and Maxillary Surgery, Central Clinical Hospital MSWiA, 02-507 Warsaw, Poland; eradzikowska@radzikowskaclinic.pl; 3Department of Urology and Urological Oncology, Multidisciplinary Hospital in Lublin, 20-400 Lublin, Poland; ewa_niezabitowska@interia.pl; 4Department of Chemistry, University of Life Sciences in Lublin, 20-950 Lublin, Poland; joanna.matysiak@up.lublin.pl; 5Department of Molecular Biology, The John Paul II Catholic University of Lublin, 20-708 Lublin, Poland; kubin@kul.pl

**Keywords:** mesenchymal stem cells, MSC-based cell therapy, quinazoline derivatives, cancer treatment

## Abstract

Mesenchymal stem cells (MSCs) are considered to be a powerful tool in the treatment of various diseases. Scientists are particularly interested in the possibility of using MSCs in cancer therapy. The research carried out so far has shown that MSCs possess both potential pro-oncogenic and anti-oncogenic properties. It has been confirmed that MSCs can regulate tumor cell growth through a paracrine mechanism, and molecules secreted by MSCs can promote or block a variety of signaling pathways. These findings may be crucial in the development of new MSC-based cell therapeutic strategies. The abilities of MSCs such as tumor tropism, deep migration and immune evasion have evoked considerable interest in their use as tumor-specific vectors for small-molecule anticancer agents. Studies have shown that MSCs can be successfully loaded with chemotherapeutic drugs such as gemcitabine and paclitaxel, and can release them at the site of primary and metastatic neoplasms. The inhibitory effect of MSCs loaded with anti-cancer agents on the proliferation of cancer cells has also been observed. However, not all known chemotherapeutic agents can be used in this approach, mainly due to their cytotoxicity towards MSCs and insufficient loading and release capacity. Quinazoline derivatives appear to be an attractive choice for this therapeutic solution due to their biological and pharmacological properties. There are several quinazolines that have been approved for clinical use as anticancer drugs by the US Food and Drug Administration (FDA). It gives hope that the synthesis of new quinazoline derivatives and the development of methods of their application may contribute to the establishment of highly effective therapies for oncological patients. However, a deeper understanding of interactions between MSCs and tumor cells, and the exploration of the possibilities of using quinazoline derivatives in MSC-based therapy is necessary to achieve this goal. The aim of this review is to discuss the prospects for using MSC-based cell therapy in cancer treatment and the potential use of quinazolines in this procedure.

## 1. Introduction

Cancer diseases are one of the major health problems all over the world. As a cell cycle disease, cancer is associated with uncontrolled cell division. Cell therapy is still evolving and seems to be a promising therapeutic approach for the treatment of many diseases, including cancer. The unique biological characteristics of Mesenchymal Stem Cells (MSCs) makes them useful in many areas such as cardiovascular diseases, liver diseases, nervous system diseases and regenerative medicine. The results of MSC therapies achieved so far have been encouraging. Searching for new therapeutic solutions in the field of cancer treatment, scientists point to the important role of MSCs in tumorigenesis and metastasis. The broad distribution of MSCs in the human body causes close mutual contact between MSCs and other types of cells, including cancer cells. Many studies have indicated that MSCs exhibit tropism for sites of tumor microenvironment and interact with cancer cells through paracrine signaling. Due to the pro-oncogenic and anti-oncogenic properties of MSCs, understanding the mechanisms of the interplay between MSCs and tumor cells is of high interest to researchers [1].The ability of MSC-secreted molecules to promote or block a variety of signaling pathways may be crucial in developing new therapeutic strategies based on MSCs. There are many anti-cancer drugs on the market, but new therapeutic solutions to overcome systemic drug toxicity and drug resistance to cancer are still being sought. It has been proven that MSCs have the ability to selectively home in on damaged tissues, tumors and metastases following systemic administration. However, it should also be noted that the differentiation potential and immunosuppressive activity of MSCs make them a key factor in tumor development [2]. Therefore, the possibilities of using MSCs as vectors for therapeutic substances are being explored. This review focuses on the benefits and barriers of using MSC cell therapy in cancer treatment. In addition, we present quinazoline derivatives as potential anticancer drugs that could be successfully used in MSC-based cell therapy. The biological and pharmacological properties of quinazoline compounds make them attractive in the selection of new substances that could be effectively delivered by MSCs to cancer cells.

## 2. General Properties of MSCs

MSCs were first isolated from adult bone marrow, described as non-hematopoietic stromal cells, in the late 1960s by Friedenstein et al. [3,4]. They were initially defined as colony-forming-unit-fibroblasts (CFU-Fs). In 1991, Caplan et al. defined CFU-Fs as “mesenchymal stem cells” (MSCs) based on their differentiation pattern [5,6]. Ultimately, MSCs are characterized as self-renewing cells with long-term viability, high migration and multi-differentiation potential [7,8,9]. According to the minimal criteria presented by the International Society for Cellular Therapy (ISCT), MSCs are cells with plastic-adherent properties in standard culture conditions and they express CD105 (known as endoglin), CD73 (known as ecto-5′-nucleotidase) and CD90 (known as Thy-1) surface markers. The expression of those surface antigens allows for rapid identification of an MSC cell population [10]. These markers have been also researched as possible therapeutic targets in cancer cells [11,12,13].

It has been found that CD105 induces activation and proliferation of endothelial cells. This protein is expressed in haematological cancers and in the vessels of solid tumors. It is worth noting that endoglin is a transmembrane glycoprotein and one of the transforming growth factor β (TGF-β) co-receptors [11]. TGF-β is a pleiotropic cytokine regulating cellular apoptosis, differentiation, migration and proliferation [14]. Ecto-5′-nucleotidase is a protein that is expressed on multiple cells and overexpressed in many cancers. CD73 mediates the suppression of immunity via adenosine, protects endothelial barrier function and inhibits leukocyte trafficking in inflammation. Hypoxia and inflammatory factors significantly influence CD73 expression. It has been reported that CD73expression is positively associated with tumor growth, angiogenesis, metastasis and poor prognosis [12,15]. It has been found that CD90 plays an important role in regulating cell adhesion, migration, apoptosis, cell-cell and cell-matrix interactions. This protein also influences axon growth, T-cell activation, and fibrosis. There is evidence that CD90 regulates cancer cell proliferation, metastasis, and angiogenesis. However, several reports also show tumor suppressor activity of CD90 [13].

MSCs express CD105, CD73 and CD90, while not expressing the markers of hematopoietic cells CD45 (a pan-leukocyte marker), CD34 (marker of primitive hematopoietic progenitors and endothelial cells), CD14 (marker of monocytes and macrophages), CD11b (marker of monocytes and macrophages), CD19 (marker of B cells), CD79α (marker of B cells) and HLA-D (marker expressed on MSC under INF-γ stimulation) [10]. In addition, MSCs have the ability to differentiate into mesodermal (adipocytes, osteoblasts, chondrocytes and myocytes), ectodermal (i.e., neurocytes) and endodermal (i.e., hepatocytes) lineages [10,16]. The main sources of MSCs are bone marrow and adipose tissue [17]. These cells are responsible for supporting hematopoietic stem cells and promoting hematopoiesis in bone marrow [18]. MSCs isolated from liposuction materials can differentiate into adipose cells, and can promote an immune-evasive environment [19]. There are increasing reports that MSCs can also be relatively easily isolated from other human tissues, including lung [20], skeletal muscle [21], peripheral blood [22], umbilical cord blood [22], amniotic fluid [23] and placenta [24]. MSCs play an essential role in these tissues, supporting their homeostasis and integrity [25,26,27]. In addition to efficient isolation from numerous tissues, MSCs can be expanded in culture without loss of functionality [28]. Moreover, they can be genetically modified with viral vectors [28,29]. The biological activity of isolated MSCs depends on local, systemic and environmental factors. Recent research has shown that the ability of umbilical cord–derived stem cells to self-renew and survive depends on the age of the mother [30]. Therefore, the origin of the isolated MSCs may determine the success of using MSCs in research and therapy.

## 3. Therapeutic Strategies Based on MSCs

Due to the traits of MSCs such as tumor tropism, deep migration into the tumor microenvironment, immune evasion, wide availability and expandability, there is a wide interest in the use of MSCs as tumor-specific therapeutic agent-loaded vehicles. MSCs have been considered as a convincing approach for drug and gene delivery in a variety of diseases and disorders, including cancer (Table 1) [31,32,33,34,35,36,37,38,39,40,41,42,43,44,45,46,47,48,49,50].

### 3.1. Interactions of MSCs with Normal and Tumor Cells

It is known that MSCs participate in the regeneration of damaged tissues and may be useful tools in regenerative medicine. Stem cell therapy may be applied in the treatment of various diseases, such as cardiovascular, digestive and nervous system diseases [1,51]. It is also believed that MSCs could be used to treat cancer, as these cells are tropic to tumor and metastatic niches and then incorporate into the tumor stroma [52,53]. The migration of these cells is determined by inflammatory signaling similar to a non-healing wound [54]. It has been reported that MSCs recruited into the tumor microenvironment may have varied potential and origin due to different types of MSCs (from distinct tissues) expressing a distinct set of genes [17,55,56]. The possibility of these cells transiting into tumor-associated fibroblasts is one of the factors contributing to oncogenesis [57]. Migration of MSCs to the tumor stroma has been demonstrated in several tumor entities, such as gastric cancer [58], pancreatic cancer [59], hepatocellular carcinoma (HCC) [60], ovarian cancer [61] and prostate cancer [62]. Kidd et al. showed that not only endogenous MSCs can migrate to the sites of inflammation, but exogenous MSCs, after systemic injection, also have this property. In the study on SCID mice with cancer, the authors demonstrated using bioluminescent imaging that, after both intravenous and intraperitoneal administration of MSCs, these cells selectively implant at the site(s) of tumor growth. Similar results were observed in the Balb/C immunocompetent syngeneic breast cancer model in which MSCs were specifically co-localized with subcutaneous neoplasms [52]. Research has also indicated that MSCs possess significant immunological functions, including immunomodulation, autocrine and paracrine activities and the evasion of innate immunity [50,63]. These cells are characterized by a low level of expression of major histocompatibility complex (MHC) I, no expression of MHC II, and no expression of costimulatory ligands (CD40, CD80 and CD86) [64]. Therefore, MSCs are considered to be poorly immunogenic, meaning that there is no need for immunosuppression during allogenic transplantation. Simultaneously, this research points out that MSCs may differentially influence the proliferation of immune cells by secreting various immunomodulators, such as human leukocyte antigen (HLA-G), indoleamine 2,3 dioxygenase (IDO), nitric oxide (NO), prostaglandin (PGE2), interleukin (IL)-6, and IL-10 [9,47]. HLA-G secreted by MSCs inhibits the migration of neutrophils to the site of injury and the generation of reactive species of O2. MSCs can affect T-cells by secreting IDO, which alters the cytokine secretion profile of naïve and effector T cells. In addition, IDO suppresses cellular or mitogen-induced T-cell proliferation. The phenotype of NK cells changes under the influence of NO produced by MSCs. NO reduces the proliferation of NK cells and influences their cytotoxic properties and modifies the set of cytokines secreted by them. Mediators secreted by MSCs also play a role in B cell proliferation. PGE2 released by MSCs inhibits the proliferation of B cells and has an effect on the chemotaxis and terminal differentiation of these cells. The production of interleukins by MSCs leads to the induction of regulatory T (TREG) cells either directly (IL-10 dependent) or indirectly (IL-6 dependent) through the generation of immature dendritic cells (DC). IL-10 secreted by MSCs promotes the expansion and function of TREG cells [65]. It is also known that MSCs are recruited to tumor sites by soluble factors secreted by cancer cells that can induce both phenotypic changes and modify the gene expression of MSCs. As a consequence, MSCs surrounding cancer cells may differentiate into more mature mesenchymal cells, and subsequently may contribute to tumor growth and metastasis. MSCs may participate in various stages of carcinogenesis through their immunosuppressive and immunomodulatory properties in the tumor microenvironment [66]. The mediators released by MSCs may inhibit apoptosis of cancer cells and enhance the promotion of cell proliferation, angiogenesis, and metastases. Cancer cell proliferation is stimulated by MSC-derived chemokines CXCL1/2 and CXCL12/SDF-1 that activate signaling pathways dependent on respective CXCR2 and CXCR4 receptors expressed by cancer cells [9]. Moreover, vascular endothelial growth factor (VEGF) and fibroblast growth factor 2 (FGF2) are believed to be two key angiogenic factors secreted by MSCs that promote tumor neovascularization [57]. MSCs have been reported to stimulate tumor growth and angiogenesis in breast and colorectal cancer [67,68]. The mechanism proposed by Huang and coworkers is that the secretion of interleukin-6 (IL-6) from MSCs increases the secretion of endothelin-1 (ET-1) in cancer cells, which induces the activation of Akt and ERK in endothelial cells, thereby enhancing their capacities for recruitment and angiogenesis to tumors [68]. However, many studies have shown that MSCs also have tumor-suppressive properties. MSCs have been reported to inhibit proliferation in pancreatic and prostate cancer [69,70]. The observations reported by Cousin and coworkers demonstrate that MSCs have the ability to interfere with the proliferation of tumor cells by altering cell cycle progression. Tumor cells under the influence of MSCs were in a resting state (G0/G1), and with down regulated CDK4 and cyclin D1 [69]. This indicates that MSCs can suppress tumorigenesis by increasing inflammatory cell infiltration [71], inhibiting angiogenesis [72], suppressing the signaling of Wnt and AKT [73], and inducing apoptosis and cell cycle arrest [74]. The ability of MSCs to inhibit capillary growth was demonstrated in a matrigel angiogenesis assay [72]. Dasari et al. reported that downregulation of XIAP (X-linked inhibitor of apoptosis protein) by treatment with cord blood MSCs induces apoptosis leading to the death of glioma cells and xenograft cells. XIAP, as one of the important anti-apoptosis family inhibitors, is upregulated in many malignancies and promotes invasion, growth, metastasis and the survival of cancer cells [74].

### 3.2. MSCs Can Affect PI3K/AKT and Wnt Signaling Pathways in the Tumor Microenvironment

MSCs can affect tumor cells through secreted extracellular proteins triggering the activation of various cell signaling pathways, in particular those related to cell growth and apoptosis regulation in cancer cells. PI3K/AKT and Wnt pathways are signaling pathways that are similar for both MSCs and tumor cells. The role of these pathways is to regulate differentiation and self-renewal potential, and determine cell fate [75,76]. The effects of MSCs on the PI3K/AKT and Wnt signaling pathways in cancer cells seem to be of particular interest. As research shows, MSCs can upregulate or downregulate the activity of these signaling pathways, and consequently they can promote or suppress both proliferation and the apoptosis of tumor cells. The phosphatidylinositol-3-kinase/protein kinase B (PI3K/AKT) signaling pathway plays an important role in regulating cell proliferations, cell apoptosis, angiogenesis and metastasis [77]. This intracellular pathway is activated by estrogen receptor (ER), IL-6 receptor (IL-6R), G protein-coupled receptors, tyrosine kinase receptors and vascular endothelial growth factor receptor (VEGFR). It has been shown that MSCs secrete paracrine molecules, such as IL-6 and VEGF, which can be ligands for these receptors. IL-6, CCL5/RANTES, SDF-1 are prooncogenic factors released by MSCs. VEGF and FGF2 are proangiogenic factors secreted by MSCs. As some studies show, CCL5/RANTES and SDF-1 may also potentially initiate signaling via the PI3K/AKT pathway [78]. Although the factors released by MSCs can upregulate the PI3K/AKT signaling, some researchers note that MSCs can also reduce the activity of this pathway. The tumor-specific antiproliferative effects of paracrine factors secreted by MSCs were observed by Yulyana et el. for hepatocellular carcinoma (HCC). The authors showed that MSCs can inhibit the growth of liver cancer because MSCs, by increasing the expression of IGFBP proteins, reduce the activation of IGF-1R/PI3K/AKT signaling [79]. Another study indicates that MSCs can inhibit the proliferation, invasion, and migration of glioma cells and promote apoptosis of these cells by downregulating the PI3K/AKT pathway. It has been shown that this is because MSCs increase the levels of proteins such as annexin A1 [ANXA1], 14-3-3 protein epsilon [YWHAE], and tumor protein p53 [TP53], and decrease the levels of proteins such as CD40, the mechanistic target of rapamycin [mTOR], and heat shock protein beta-1 [HSPB1] in glioma cells [80].

The Wnt signaling pathway is one of the key cascades regulating cell survival, differentiation, morphogenesis and proliferation. Wnt activators are secreted Wnt proteins (a family of 19 secreted glycolipoproteins) that work in an auto-, para- and endocrine manner. Disruptions in Wnt signaling have been found in many pathophysiological states, including different types of cancer [81]. Wang et al. showed that MSCs can promote metastatic growth and the chemoresistance of cholangiocarcinoma cells via the Wnt/β-catenin signaling pathway. MSCs promote β-catenin nuclear translocation and up-regulate the Wnt target genes’ MMPs family, cyclin D1 and c-Myc. In addition, the researchers indicated that β-catenin is related to Pathologic Tumor-Node-Metastasis (PTNM) and may be a key factor in the development of cholangiocarcinoma. The study suggests that MSCs can increase β-catenin expression and stimulate Wnt activity by MSC-secreted soluble factors and the previously mentioned molecular mechanisms [82]. On the other hand, there is research that underscores the importance of MSCs in inhibiting the proliferation, invasion, and migration of cancer cells through the Wnt signaling pathway [83,84]. It has been reported that MSC-derived exosomes containing miR-133b suppress glioma development by inhibiting EZH2 and the Wnt/β-catenin signaling pathway [83]. Recent studies have also shown that exosomes derived from MSCs may play a significant role in cancer control. Jia et al. found that adipose derived MSC-exosomes containing miR-1236 increase the sensitivity of breast cancer to cisplatin with the involvement of SLC9A1 downregulation and Wnt/β-catenin inactivation [84]. Torsvik et al. indicate that tumor growth can also be inhibited by MSC secreted inflammation cytokines, by Dkk-1 and the subsequent inhibition of the Wnt/β-catenin pathway [78]. The cellular and molecular mechanisms of action of MSCs in the tumor microenvironment underlie the supportive and suppressive effects of MSCs on tumor growth.

However, the mechanisms of MSCs secreting molecules that promote or inhibit carcinogenesis have not been elucidated so far. Therefore, the role of recruited MSCs in tumor sites is still under discussion. A comprehensive understanding of the effects of MSCs on cancer cells is crucial for the development of treatment strategies using MSCs as vehicles of anticancer agents. Nevertheless, the results of research on MSC-based cell therapy are promising.

### 3.3. The Use of MSCs as Vehicles of Various Therapies

One of the strategies of MSC-based cell therapy is the use of MSCs as direct carriers of therapeutic agents [9,60]. The ability to load into MSCs and then to release into the tumor microenvironment has been investigated with many known and used drugs, such as doxorubicin, gemcitabine, paclitaxel and sorafenib (Table 1). As studies show, the loading capacity of MSCs for gemcitabine and paclitaxel was evaluated most highly. In addition, it has been found that hydrolipophilicity plays a role in drug release by MSCs. It was noticed that Paclitaxel is characterized by higher lipophilicity, therefore it is released by MSCs in a smaller amount, while in relation to the more water-soluble gemcitabine, its higher release by MSCs was observed [85]. It is also worth noting that MSCs can be loaded with an active drug that will be released in the tumor microenvironment or an inactive form of the drug that will be activated when MSCs reach the tumor site. For instance, MSCs have been used to increase the specificity of macromolecule G114, a thapsigargin-based prostate specific antigen (PSA)-activated prodrug in prostate cancer. These cells delivered inactive prodrugs that were activated in the tumor by tumor-specific proteases such as prostate-specific antigen (PSA) or prostate-specific membrane antigen (PSMA) [86,87].

Researchers tracked the anticancer agents in MSCs’ membrane and cytoplasm. It seems that there are three mechanisms of drug loading into MSCs: (1) simple diffusion based on the chemical nature of anticancer agent, (2) endocytosis, and (3) transport of the drug by hCNT1 and hENT1 transporters [85]. Some types of therapeutic agents such as paclitaxel were found in their place of action such as microtubule networks and centriole, in mitochondria where paclitaxel is metabolized, in the Golgi apparatus, as well as in Golgi-derived vesicles of MCSs. It was also observed that anticancer agents may interfere with the normal gene expression pattern of MSCs. Researchers noted the presence of multivesicular structures with the drug originating from cell membrane budding. It seems that MSCs initiate to form exosomes near cellular membranes [88,89,90]. The secreted vesicles from paclitaxel-loaded MSC were observed, whereas no interactions with the plasma membrane, such as a gap junction or junction structure between the MSCs and cancer cells, were noticed. Simultaneously, a significantly antineoplastic effect against ductal pancreatic adenocarcinoma cells was observed [91,92]. These observations suggest that the extracellularvesicles (EVs) are recruits for transporting anticancer agents between MSCs and tumor cells [93,94].

Figure 1 shows how a small-molecule anticancer agent can be loaded into the MSC and released into the tumor microenvironment.

The use of MSCs as vehicles of anticancer agents is an opportunity to minimize the unexpected side effects of chemotherapeutic drugs and to improve clinical outcomes. MSCs can release anticancer agents in a time-dependent manner. This makes them an excellent tool for the development of therapy in which drugs will be slowly released locally to ensure an effective concentration in the tumor site. However, the release capacity depends on both the MSCs and the properties of the therapeutic agents [85]. Therefore, it is necessary to determine the best tissue source for MSC isolation and to define the route of administration of MSCs that will achieve the desired therapeutic goal. Another way to increase the possibilities for developing effective therapies using MSCs is through genetic engineering. In this type of approach, native MSCs are isolated and expanded from autologous or allogeneic donors, and then the MSCs are transfected with a therapeutic transgene (Table 1). There are two common strategies for the use of transgenes. Transgenes can be used to enable MSCs to secrete therapeutic proteins that affect cancer cells either directly or indirectly. Another strategy is to use transgenes encoding a “suicide gene”. In this case, the MSCs are genetically modified to convert a non-toxic prodrug (administered systematically) to a toxic and active form at the tumor site. After the conversion/uptake of the toxic drug, not only cancer cells but also MSCs are killed [9]. This strategy can overcome the limitations of MSC-based cell therapy, especially those related to the tumor promoting actions of MSCs. The use of genetic engineering techniques can ensure not only the effectiveness but also the safety of the treatment.

MSCs are being genetically modified to convert inactive drugs, overexpress immunomodulatory molecules and induce cancer cells apoptosis. For instance, MSCs have been genetically manipulated to express specific enzymes to convert inactive systematically administered prodrugs like ganciclovir and fluorouracil (5-FU) into active cytotoxic agents [95,96]. This treatment approach improves tumor-specificity and reduces the side effects of systemic toxicity. MSCs can also be genetically engineered to overexpress selected immunomodulatory cytokines that promote the action of killing cancer cells. This result was obtained in melanoma cells in which MSCs genetically modified to produce IFNβ induced a significant anti-proliferative effect [97]. Lu et al. showed in animal models that adipose MSCs combined with a modified E7′ antigen possess an antitumor effect in colon and lung cancer by inducing cell apoptosis [98]. In another example of cancer gene therapy strategy, adipose MSCs have been modified to release a soluble trimeric and multimeric variant of known anti-cancer TNF-related apoptosis-inducing ligand (sTRAIL) soluble anti-cancer factors, which could lead to apoptosis in the cell lines of primary pancreatic ductal adenocarcinoma [99]. Overall, genetic engineering capabilities have resulted in both the adaptive and innate immune systems being successfully stimulated to inhibit cancer growth [9].

Nevertheless, the challenge remains to match the type of therapy to the tumor type in order to obtain the best possible treatment results. MSC-based cell therapy also has several other limitations, including non-specific dissemination throughout the body and ineffective local concentration of therapeutic agents in cancer niches [100]. Moreover, the potential of MSCs to differentiate into mesenchymal lineages may increase immunogenicity and reduce the therapeutic effects [101]. It seems that the best way to avoid tumor-supporting effects is to engineer MSCs containing anticancer therapeutics so that they become inactive or die when released [9]. Niess et al. found that the application of CCL5/HSV-TK transfected MSCs in combination with ganciclovir significantly reduced HCC growth. The researchers discovered that exogenously added MSCs are recruited to the tumor site with concomitant activation of the CCL5 or Tie2 promoters within MSCs. In the case of treatment of experimental HCC, stem cell-mediated delivery of suicide genes into the tumor followed by prodrug administration was effective [60]. Another promising solution is a therapeutic approach using MSC exosomes. Altanerova et al. demonstrated that MSCs expressing the yeast cytosine deaminase::uracil phosphoribosyl transferase (yCD::UPRT) suicide fusion gene and labeled with an iron oxide carbohydrate nanoparticle released exosomes that contained iron oxide. The researchers indicated that MSC exosomes packaged with magnetic nanoparticles can be efficiently endocytosed by cancer cells. In this study, exosomes from labeled MSCs expressing the suicide fusion gene in the presence of the prodrug 5-fluorocytosine suppressed tumor growth. In addition, magnetically induced hypothermia facilitated the targeted ablation of cancer cells [102].

Although a deeper understanding of the cellular and molecular interactions between MSCs and the tumor microenvironment is necessary, the immune privilege, plasticity and migratory potential of MSCs, have made them ideally suited for cellular-based immunotherapy and as vehicles for gene and drug delivery in a wide range of diseases and disorders, including cancers (Table 1) [9,28].

## 4. Are anticancer Quinazoline Derivatives Good Candidates to Be Delivered with MSCs?

One of the potential chemotherapeutic agents that can potentially be delivered by MSCs to tumor sites are quinazoline derivatives. The importance and application of these compounds and MSCs in cancer therapy has so far been independently investigated, but the results of these studies are encouraging in both subjects. Quinazolines are a group of compounds whose ability to be loaded and released by MSCs has not yet been of significant interest to scientists. However, research on other anti-cancer agents, on MSCs which have an ability to tropic tumor site, and the discovered mechanisms of small-molecules anticancer agents uptake by MSCs and their release into the tumor microenvironment support the possibility of using quinazoline derivatives in MSC-based cell therapy. To confirm this, research is needed to reveal the existing interactions between quinazolines and MSCs, and to determine the ability of MSCs to load and release these small-molecule anticancer agents.

Quinazolines are one of the most active classes of nitrogen containing heterocyclic compounds. A quinazoline structure includes two fused six-membered simple rings (**1**) [103] (Figure 2). Derivatives of quinazoline, quinazolinones, are the oxidized form of quinazoline (ketoquinazoline). Depending on the position of the keto (oxo) group, they may be classified into three types: 2(1H)-quinazolinones (**2**), 4(3H)-quinazolinones (**3**) and 2,4(1H,3H)-quinazolinedione (**4**) (Figure 2).

These rings are also part of quinazoline alkaloids (**5**–**7**) (Figure 3). Among these three quinazolinone structures, 4(3H)-quinazolinones are more prevalent, either as intermediates or as natural products in many proposed biosynthetic pathways (**5**) (Figure 2).

Quinazolines and quinazolinones occur naturally and artificially [104]. Currently, derivatives of quinazoline are widely used in clinical practice, and the methods of their synthesis are still developing due to the potential variety of their medical uses. They have a significant and wide range of pharmacological activity which is associated with the presence of two fused six-membered simple aromatic rings, a benzene and a pyrimidine ring as well as heteroatoms [103,104]. The numerous biological properties of quinazolines have raised great interest in the scientific community. Among these properties are antibacterial, anti-tuberculosis, antimalarial, antifungal, antiviral, antihypertensive, antioxidant, analgesic, anticonvulsant, anti-inflammatory and antitumor activities [105,106,107,108,109,110,111,112,113]. A particular field of medical application for quinazolines appears to be in the treatment of cancer. In recent years, The Food and Drug Administration (FDA) has approved several quinazoline derivatives as a new class of cancer chemotherapeutic agents with significant therapeutic efficacy against solid tumors. Anticancer drugs such as gefitinib (**8**), erlotinib, lapatinib (**9**), afatinib (**10**), and vandetanib were approved for clinical use (Figure 3) [114]. However, the potential of quinazolines is still being researched, and new therapeutic approaches for their use are also being sought. It has been shown that quinazoline derivatives can exhibit antitumor effects via multiple ways, for instance as an inhibitor of phosphatidylinositol-3-kinase (PI3K) and an inhibitor of receptor tyrosine kinases (RTK) [115,116]. Tyrosine kinases (TKs) play a key role in signal recognition, transduction and amplification. It has been established that many cellular processes require tyrosine phosphorylation. However, loss of control over cell proliferation, differentiation and migration can lead to serious diseases, including the onset of cancer, and its progression and metastasis [117,118,119]. Some of the new TK inhibitors are shown in Figure 4 (**11**–**13**). They include pyrrole (**11**), morpholine(**12**) or urea (**13**) moieties and were designed on the basis of gefitinib or other quinazoline anticancer drugs being TK inhibitors (Figure 4) [120,121]. Quinazoilnes can also act as tubulin inhibitors on cell lines and can lead to apoptosis (**14**–**16**) (Figure 3) [122,123,124,125]. Verubulin (derivative **14**) had reached phase II of a trial for the treatment of glioblastoma, but it was halted due to cardiovascular toxicity. Recently, Li et al. described a new potent tubulin inhibitor with a pyridine ring (**16**) showing nanomolar activity against some cancer cell lines (Figure 3) [126].

Tubulin, as the main structural protein of microtubules, is important for cell division and motility. Therefore, tubulin is a target for anticancer drugs such as vincristine, docetaxel, paclitaxel and others. These agents prevent the formation of a mitotic spindle by binding to tubulin. Another effect of compounds that inhibit tubulin is the ability to inhibit cell migration and invasiveness. Thus, these agents prevent metastases in neoplastic diseases [125,126]. Some compounds act as multifunctional ligands. Fluorinated aminoacid quinazoline hybrids were described as dual inhibitors of 650 tubulin polymerization and epidermal growth factor receptor (EGFR) kinase **17** [127]. Compound **18** is a cyclin-dependent kinase 4 (Cdk4) and tubulin 653 polymerization inhibitor (Figure 5) [128].

The inhibitory effects of quinazolines on voltage-dependent sodium channels (VDSCs) can lead to antiangiogenic and analgesic effects [129]. VDSCs are not only expressed in excitable cell systems. High expression of these channels is also observed in metastatically active cells. Therefore, VDSCs are considered as a promising target for antitumor and antimetastatic therapy [130,131]. Pyrrolidinyl carbamate derivatives of quinazoline were described and patented as VDSC inhibitors **19** (Figure 5) [132]. Other quinazoline-based anticancer agents can act as protein lysine methyltransferase inhibitors, including diazepam and piperidyne rings **20** [133], or as PI3K/Akt/mTOR inhibitors **21** (with morpholine andnicotinonitrile moieties) (Figure 5) [134].

Another ability of quinazolines is to overcome multidrug resistance (MDR) in tumors by inhibiting ATP-Binding cassette (ABC) transporters. The overexpression of ABC transport proteins such as ABCB1 or ABCG2 causes the efflux of harmful substances out of cells at the cost of ATP hydrolysis. Inhibition of the ABC transporters by potent and selective inhibitors may increase the efficacy of cancer therapy. The expression of ABC transporters is often associated with the occurrence of MDR in chemotherapy of multiple cancers [135,136]. Quinazolines have also been shown to exhibit satisfactory cytotoxic activity and can significantly inhibit the proliferation of different human cancer cell lines such as MCF-7, BT-474 and MDA-MB231 human breast cancer cell lines, the A-549 human lung adenocarcinoma cell line, the HeLa cervical cancer cell line, the HT-29 human colorectal adenocarcinoma cell line, the A375 melanoma cancer cell line and the D283 medulloblastoma cell line [114,137,138,139,140].

Currently, it is also indicated that multifunctional hybrid compounds containing quinazoline/quinazolinone and other biologically active pharmacophores can potentially significantly increase the effectiveness of pharmacotherapy [141]. Osipov et al. point out that hybrid compounds containing the quinazoline cycle may constitute the basis for the synthesis of effective drugs with great therapeutic potential in the complex treatment of oncological, neurological or infectious diseases [142]. There are many reports that show the therapeutic potential of various quinazoline/quinazolinone-based hybrid analogues against a wide range of diseases. Additionally, the properties of the newly synthesized quinazolines are still being investigated, and the results are promising.

## 5. Conclusions and Perspectives

In light of the proven properties of quinazoline derivatives in relation to different cancer cell lines, they seem to be an attractive choice for new therapeutic solutions in the treatment of neoplasms and metastases. The drugs currently used in cancer diseases give hope that the synthesis of new quinazoline derivatives and the development of new methods of their application may contribute to the establishment of effective and low-toxic therapies for oncological patients. MSCs seem to be a promising therapeutic tool to achieve this goal. Systematically applied conventional cancer treatment agents are characterized by the lack of tumor specificity. They cause insufficient concentration of the chemotherapeutic agent in tumor cells and toxic effects in normal cells. These obstacles in cancer treatment can be overcome by using the migratory capacity of MSCs. A more accurate characterization of the subtypes of MSCs and a better understanding of the specific molecular mechanisms underlying their pro-tumorigenic properties could enable the development of highly effective therapeutic strategies based on MSCs. There is also a need to find therapeutic drugs that can be successfully used in this type of therapy. Quinazolines are potential candidates for this purpose due to their biological and pharmacological anticancer properties. Therefore, researching and understanding the interactions between MSCs and quinazoline derivatives may show new perspectives in cancer treatment. The use of MSCs as a delivery vehicle of small-molecule anticancer agents with activities such as quinazolines may turn out to be a milestone in the treatment of cancer patients.

## Figures and Tables

**Figure 1 ijms-23-02745-f001:**
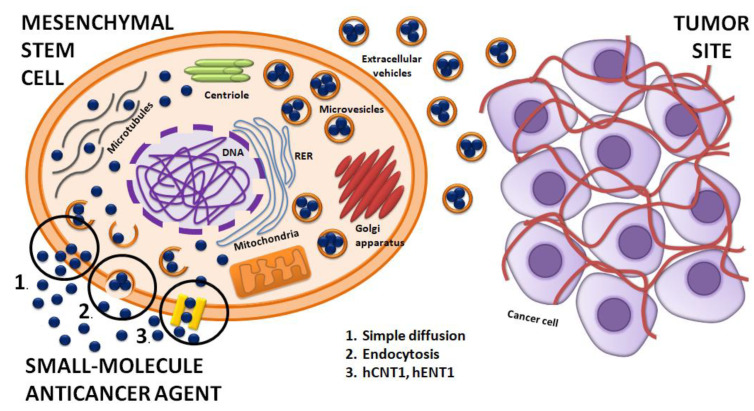
Therapeutic agent delivery by MSC to the tumor site. Mechanisms of small-molecule anticancer agent loading into MSC: 1. Simple diffusion; 2. Endocytosis; 3. Transporters: hCNT1, hENT1. MSC produce vesicles that contain small-molecule anticancer agents. The presence of extracellular vesicles (EVs) between MSCs and cancer cells suggests that the small-molecule anticancer agent can be delivered to cancer cells in a vesicular system [84].

**Figure 2 ijms-23-02745-f002:**
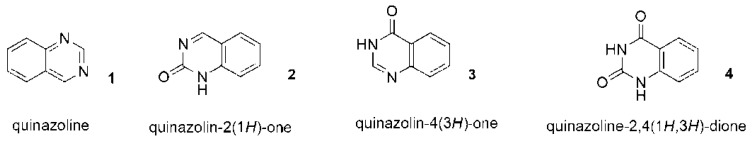
(**1**) Structure of quinazoline; (**2****–4**) Structures of quinazolinones.

**Figure 3 ijms-23-02745-f003:**
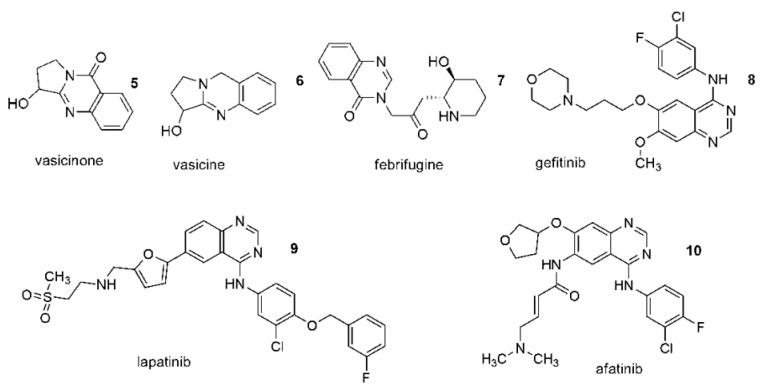
(**5****–7**) Quinazoline based alkaloids; (**8****–10**) Anticancer drugs.

**Figure 4 ijms-23-02745-f004:**
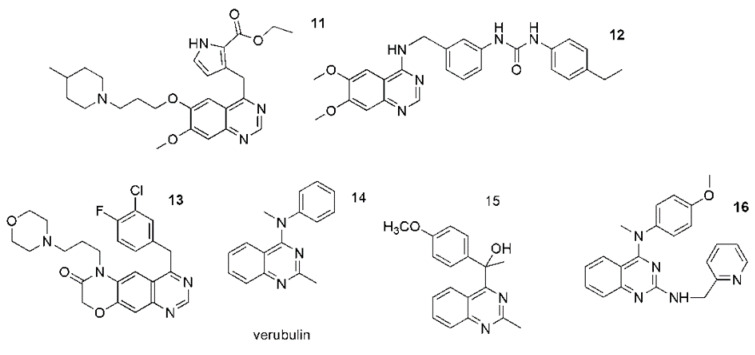
Quinazoline derivatives as (**11**–**13**) tyrosine kinase receptor inhibitors; (**14**–**16**) Tubulin inhibitors.

**Figure 5 ijms-23-02745-f005:**
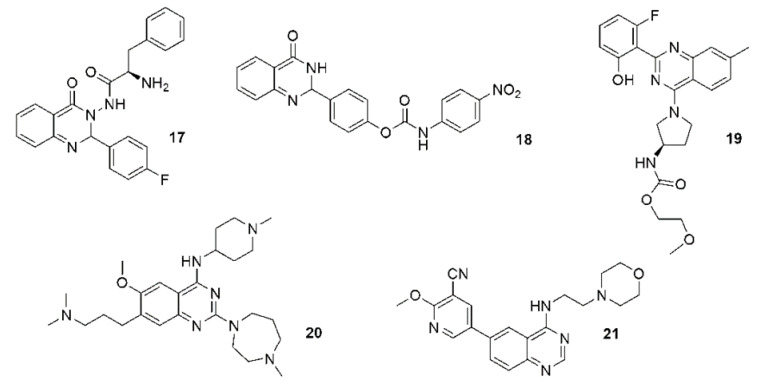
(**17**–**21**) Quinazoline based anticancer agents of various molecular targets.

**Table 1 ijms-23-02745-t001:** Examples of the use of MSCs as carriers for drug and gene loading described in the literature [31,32,33,34,35,36,37,38,39,40,41,42,43,44,45,46,47,48,49,50].

Loaded Drug/Expressed Transgene	Target Disease	Experimental Model	Therapeutic Effect	Study
TNF-related apoptosis-inducing ligand (TRAIL)	Glioma	Glioblastoma cells (C6)	Apoptosis of tumor cells	Tang X.J. et al. [31]
Galectin-1	Allergic Airway Disease (AAD)	Mouse model	Anti-inflammatory effect	Ge X. et al. [32]
Doxorubicin (DOX)	Colorectal cancer	Female BALB/c mice (4–6weeks), C26 and MCF7 cell lines	Significant tumor growth inhibition in comparison with free DOX	Bagheri E. et al. [33]
Gemcitabine	Pancreatic cancer	Human pancreatic carcinoma (pCa) cells	Growth inhibition of a human pCa cell line in vitro	Bonomi A. et al. [34]
Ptx-PLGA NPs	Glioma	Rat model	Tumor cell death, prolonged survival	Wang X. et al. [35]
Paclitaxel (PTX)	Melanoma lung metastasis	Syngeneic murine model	Inhibition of the formation of lung metastasis	Pessina A. et al. [36]
Paclitaxel (PTX)	Pancreatic cancer	Human pancreatic cell line CFPAC-1	Strong anti-proliferative effect	Pascucci L. et al. [37]
Interferon-β (IFN-β)	Ovarian cancer	Syngeneic mouse tumors (ID8-R) and human xenograft (OVCAR3, SKOV3) tumor models	Modulation of tumor kinetics resulting in prolonged survival	Dembinski J.L. et al. [38]
(C-X3-C motif) ligand 1 (CX3CL1)	Light-induced retinal degeneration	Rat model	Neuroprotective and immunomodulatory effects	Huang L. et al. [39]
Multineurotrophin MNTS1	Spinal Cord Injury (SCI)	Rat model	Promotion of cell growth and improvement of sensory function after SCI	Kumagai G. et al. [40]
GATA binding protein 4 (GATA-4)	Myocardial infarction	Cardiomyocytes	A significant increase the number of blood vessels, a decrease the proportion of apoptotic cells, and an increase the mean number of cardiac c-kit-positive cells	He J.G. et al. [41]
Interleukin (IL)-18	Breast cancer	Mouse model	Inhibition of tumor cell proliferation and tumor angiogenesis, induction of a more pronounced and better therapeutic effect at tumor sites, especially in early tumors	Liu X. et al. [42]
Pigment epithelium-derived factor (PEDF)	Glioma	Mouse model	Apoptosis of glioma cells and prolonged the survival	Wang Q. et al. [43]
C-X-C chemokine receptor type 4 (CXCR4)	Inflammatory bowel disease (IBD) and IBD-induced cancer	Mouse model	Anti-tumor effect	Zheng X.B. et al. [44]
Glial cell line-derived neurotrophic factor (GDNF)	Parkinson’s disease	Rat model	Localized neuroprotective effect	Hoban D.B. et al. [45]
Angiotensin-converting enzyme 2 (ACE2) gene	Lipopolysaccharide-Induced Lung Injury	Mouse model	Improvement of the lung histopathology; anti-inflammatory effects; reduction of pulmonary vascular permeability; improvement of endothelial barrier integrity, and normalization of lung eNOS	He H. et al. [46]
Brain-derived neurotropic factor (BDNF) gene	Severe Neonatal Hypoxic Ischemic Brain Injury	Rat model	Supression of the increase in cytotoxicity, oxidative stress, and cell death in vitro; significant attenuating effects on severe neonatal HI-induced short-term brain injury scores, long-term progress of brain infarct, increased apoptotic cell death, astrogliosis and inflammatory responses, and impaired negative geotaxis and rotarod tests in vivo	Ahn S.Y. et al. [47]
Oxidation Resistance 1 (OXR1) gene	Immune-mediated nephritis	Mouse model	Protective effect on nephritis by suppressing inflammation and oxidative stress	Li Y. et al. [48]
Insulin-like growth factor-1 (IGF-1)	Chronic Chagas disease	Mouse model	Immunomodulatory and proregenerative effects to the cardiac and skeletal muscles	Silva D. N. et al. [49]
Human tissue kallikrein (TK) gene	Cardiac injury	Rat model	Protect against cardiac injury, apoptosis and inflammation, and promote neovascularization to improve cardiac function	Gao L. et al. [50]

## Data Availability

Not applicable.

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
