# Peer review of "MSCs as Tumor-Specific Vectors for the Delivery of Anticancer Agents—A Potential Therapeutic Strategy in Cancer Diseases: Perspectives for Quinazoline Derivatives"

_ijms, 2022, doi:10.3390/ijms23052745_

Round 1

Reviewer 1 Report

For me, there is a mismatch between the title and the content of the review.
The title suggests that the available data are much more complete and advance than what currently exists.
The review is actually divided in three separate parts
1) A review on the role of MSCs in cancer with pro and antitumoral functions.
2) A review on the potential use of MSC (or exosome) as vector to deliver drug
3) A review on quinazoline.

There is almost no link between the three parts.
1) Part one is a classical review on MSCs in cancer, interesting and well written but not original, several review exist on the subject
2) This part is the most original and could be developed as the real subject of the review. The authors could integrate in this part a short analyze of the pro or antitumoral effect of MSC and how to use it or manipulate it when you use MSCs as vehicle. Also discussing more the use of MSC compared to their exosomes.
3) The last part describes quinazoline family as potential chemotherapies agents but the link with MSC is still totally speculative since there are no preliminary data to support it.
So it could be discussing as future perspective but I don't think that it fit in this review as a full paragraph suggesting that it is something already tests and work in progress.

In conclusion, I think the authors should probably split the review in two:
one on MSCs in cancer and one on MSCs as vector/vehicle for targeted therapy. The second one is the most interesting and could be more developed. The part on quinazoline is a little bite premature to be fully asociated with MSCs in a review.

Author Response

We thank the Reviewer for the valuable suggestions. We have addressed them in the new version of the manuscript.

Firstly, in order to avoid a mismatch you mentioned we have modified the title and the structure of the review.

Now the chapter on therapeutic use of MSCs is a core part of the review, and we have added the issues you mentioned in your report.

We fully agree that the link between quinazolines and MSCs is speculative. That is why we now write about quinazolines as an interesting perspective and we have separated the part of quinazolines from the main part of MSCs. However we decided to combine these two issues due to the excellent anticancer properties of quinazoline derivatives and their physico-chemical characteristics that make the compounds reasonable candidates for MSC-mediated delivery.

Reviewer 2 Report

-The authors are recommended to explain the MSC-mediated tumorigenicity that related to MSC quality.

-The table is so simple and not informative and need substantial changes.

-The authors need to discuss the detailed methods of MSC-based drug delivery.

-The limitations of MSC-based delivery as an anti-cancer strategy are need to be discussed with recommendations of the further improvements.

Author Response

We thank the Reviewer for the valuable suggestions. We have addressed them in the new version of the manuscript.

We have also completely changed the table, and now it is much more detailed.

Round 2

Reviewer 1 Report

The new structure of the paper improves significantly the common thread for the readers. The link between MSCs , their potential role as vehicules for drug delivery and the potential use of quinazoline derivatives in this setting is now a clear cascade well explain in the review

Reviewer 2 Report

I recommend accepting the manuscript at the present form.